# Effect of Treating Acid Sulfate Soils with Phosphate Solubilizing Bacteria on Germination and Growth of Tomato (*Lycopersicon esculentum* L.)

**DOI:** 10.3390/ijerph18178919

**Published:** 2021-08-25

**Authors:** Jae Hwan Kim, So-Jeong Kim, In-Hyun Nam

**Affiliations:** 1Department of Human Environment Design, College of Science, Cheongju University, Cheongju 28503, Korea; smileeye77@cju.ac.kr; 2Geologic Environment Research Division, Korea Institute of Geoscience and Mineral Resources (KIGAM), Daejeon 34132, Korea; sojkim86@kigam.re.kr

**Keywords:** acid sulfate soil, phosphate solubilizing bacteria, *Methylobacterium* sp. PS, *Caballeronia* sp. EK, vegetation promotion, biofertilizer

## Abstract

Acid sulfate soils contain sulfide minerals that have adverse environmental effects because they can lead to acidic drainage and prevent the establishment of vegetation. The current study examined the effect of a novel method for the restoration of these soils and the promotion of germination and plant growth. Thus, we isolated two strains of phosphate solubilizing bacteria, *Methylobacterium* sp. PS and *Caballeronia* sp. EK, characterized their properties, and examined their effects in promoting the growth of tomato plants (*Lycopersicon esculentum* L.) in acid sulfate soil. Compared with untreated control soil, treatment of acid sulfate soils with these bacterial strains led to increased seed germination, growth of plants with more leaves, and plants with greater levels of total-adenosine tri-phosphate (tATP). Relative to the untreated control soil, the addition of *Caballeronia* sp. EK led to a 60% increase in seed germination after 52 days, growth of plants with more than 3 times as many leaves, and a 45.2% increase in tATP after 50 days. This strain has potential for use as a plant biofertilizer that promotes vegetation growth in acid sulfate soils by improving the absorption of phosphorous.

## 1. Introduction

Acid sulfate soils occur throughout the world, and they can originate from sedimentary processes, volcanic activity, or metamorphism. These soils are not suitable for agriculture and generally have adverse environmental effects because of the increased acidity caused by oxidation [1,2,3]. These soils have abundant pyrite (FeS_2_), which is associated with high acidity and the release of aluminum (Al) following oxidation by exposure to the atmosphere [4,5]. Under acidic soil conditions, Al^3+^ restricts the growth of plant roots by inhibiting cell division, cell elongation, or both [6]. Furthermore, rice plants grown on acid sulfate soils are susceptible to Fe^2+^ toxicity [7]. Thus, an acid sulfate soil is generally unsuitable for agriculture unless it is properly treated by removal of the high concentrations of aluminum and iron and its fertility is increased [8].

There are only a few acid-tolerant plant species and microbes that can survive in these soils [9]. Thus, these soils generally contain few microorganisms, and the number varies considerably depending on the vegetation type and soil management practices [2]. In addition to chemical treatments, addition of microbes may improve nutrient availability in these soils, especially by increasing the level of phosphorus and reducing Al toxicity [10]. Thus, acid sulfate soils, which typically have a pH of 3 to 5 and contain Al and/or Fe, are considered to be toxic [10], chemically degraded, and unfit for agriculture unless they are treated with appropriate amendments [11]. A surface coating technique that prevents sulfide minerals from contacting oxygen is a general method that can prevent the oxidation of sulfide minerals and inhibit the generation of acid drainage [12]. In addition, acid drainage water that is generated deep in the soil can rise to the surface by capillary action and destroy plants growing at the surface due to diseases such as blight [1,3,4].

Phosphate solubilizing bacteria (PSB) are microorganisms that supply soluble phosphorus (P) that can be used as a nutrient by plants [13,14,15]. P is an essential nutrient for plants because it is necessary for the production of nucleic acids, phospholipids, phytates, and other important compounds. P is most often present in young plants and in the growing regions of plants, where cell division is most active [13,14]. An insufficiency of P can decrease a plant’s ability to synthesize RNA and protein, thus deteriorating the plant’s nutritional status and inhibiting the growth of roots and stems [16]. Soil P is often present in an insoluble form, and it binds with the aluminum ion in acidic soils and with the calcium ion in alkaline soils, forms that plants cannot assimilate [17]. However, certain microbes can solubilize this insoluble phosphate by forming chelating substances, such as 2-ketogluconate, ferrous ion precipitation into FeS by H_2_S production, and generation of sulfuric and nitric acid by oxidation of sulfur or ammonia [18]. Furthermore, the density of soil PSB correlates with the amount of solubilized phosphate. This suggests it may be possible to achieve efficient P solubilization by increasing the activity of PSB in the soil. Thus, one approach for the remediation of acid sulfate soils is the addition of PSB [19,20]. PSB are widely distributed in soils and often occur in the rhizosphere. There are PSB in multiple genera, including *Acinetobacter*, *Alcaligenes*, *Arthrobacter*, *Azospirillum*, *Bacillus*, *Burkholderia*, *Enterobacter*, *Erwinia*, *Flavobacterium*, *Pantoea*, *Pseudomonas*, *Rhizobium*, *Trichoderma*, and *Serratia* [13,16,17,18,19].

Some previous studies have examined the development of biofertilizers, in which specific microorganisms were added to soils to increase vegetation cover. However, research and the commercial availability of these treatments are limited due to limited experience in microbial cultivation processes, the need to select suitable microbial strains, and the necessity for extensive testing [20]. Thus, there is increasing interest in the development and application of biofertilizers that can efficiently solubilize phosphate and supply it to plants because of strengthening environmental regulations and the broad demand for increasing vegetation cover. Therefore, the current study examined the effect of adding PSB to acid sulfate soils to promote plant growth and the potential of this method to facilitate the restoration of plant ecosystems.

## 2. Materials and Methods

### 2.1. Isolation and Characterization of PSBs

Plants growing near Daedeok stream (Daejeon, Korea) were collected for the isolation of pure cultures of PSB that were growing in the rhizosphere. The roots of the collected plants were cut and suspended in 0.85% NaCl to obtain samples. To collect insoluble inorganic phosphate, calcium phosphate was washed with distilled water, and the insoluble residue remaining on the filter was collected after passing it through a 0.45 μm filter; this residue was suspended in artificial freshwater (AFW) so the final concentration was 0.5%. Then, 200 μL aliquots were transferred into each well of 96 well plates. Each liter of AFW had 0.5 g KCl, 1 g NaCl, 0.5 g NH_4_Cl, 0.10 g CaCl_2_·2H_2_O, and 0.4 g MgCl_2_·6H_2_O. This solution was then sterilized at 121 °C for 15 min, and then glucose, lactate, acetate, formate, or pyruvate (50 mM each) were added as substrates. Then 1 μL of the plant root suspension was inoculated into the PSB culture medium, and was maintained in stationary culture at 25 °C for 2 weeks. Liquid cultures were performed using 5 different substrates and species of plants collected from the field, and then 5 μL of each culture was inoculated on Pikovskaya’s agar medium (Merck, Darmstadt, Germany) with yeast extract and *D*-glucose as substrates. The cells having phosphate solubilizing activity were selected by halo on Pikovskaya’s agar. Among the samples, those from reed roots had the highest phosphate solubilizing activity. To establish pure cultures, each isolated strain was subcultured repeatedly in R2A agar (BD Difco) plates, and a single colony was then obtained. The colony was cultured in R2A agar for 3 days. The genomic DNA of isolates was extracted using DNeasy Blood and Tissue kit (Qiagen). PCR was performed using 27F (5′-AGAGTTTGATCMTGGCTCAG-3′) and 1492R (5′-GGTTACCTTGTTACGACTT-3′) [21] primer set to amplify 16S rRNA gene of the extracted genomic DNA from isolates. One μL of the bacterial DNA was amplified in a 20 μL reaction mixture with 10 μL Solg™ 2X EF-Taq PCR Smart mix 1 (Solgent, Daejeon, Korea), 1 μL forward primer (27F, 10 pmol), 1 μL reverse primer (1492R, 10 pmol), 7 μL distilled water. The PCR conditions consisted of a pre-denaturation for 5 min at 94 °C, followed by 30 cycles of 94 °C for 30 s, 55 °C for 30 s (annealing), and 72 °C for 45 s (extension); and then reaction at 72 °C for 5 min (final extension). Each strain was cultured in R2A medium for 2 days to assess the effects of individual culture conditions (temperature, pH, and NaCl concentration). For these experiments, the effect of temperature was determined by growth on R2A agar at different temperatures; the effect of pH was determined by measuring the absorbance at 600 nm (OD_600_) after incubating in R2A broth from pH 4.0 to 11.0; and the effect of NaCl was determined by growth in R2A broth (MB cell) with NaCl content from 0% to 10%. Strains were characterized using API 20NE and API ZYM kits (bioMérieux) following the manufacturer’s instructions.

### 2.2. Soil and Organic Materials Properties

The acid sulfate soil samples were collected from an abandoned pyrophyllite mine in Busan, Korea (35°20′16″ N, 219°07′53″ E) that was enriched with various sulfide minerals. Soil samples were collected at a depth of 0 to 20 cm, corresponding to the common depth of plowing, because these soils are to be used as arable land in the future. Where it was applied, the depth of the topsoil cover was also at least 20 cm. Soil organic matters were determined by potassium dichromate method, and soil pH and EC were determined using a multi-meter (Orion Star A215, Chelmsford, MA, USA) in water with 1:2.5 (soil:water). Soil exchangeable cations were determined by flame atomic absorption spectrometry (ASS PerkinElmer 1100B), and P was analyzed by calorimetrically (molybdenum blue) method. Measurements of this acid sulfate soil confirmed the pH was 3.4 and the available *P* was 38.96 mg/kg (Table 1).

In addition, the chemical properties of the organic supplement used for germination and plant growth experiments were determined (Table 2). This organic supplement (cocopeat 68.0%, peatmoss 15.0%, perlite 7.0%, vermiculite 6.0%, and zeolite 4.0%) had physical properties similar to soil, was rich in nutrients, and had a pH of 7.2, indicating it provided appropriate conditions for germination and plant growth.

### 2.3. Plant Growth Experiments

Plant growth experiments were performed using tomato (*Lycopersicon esculentum* L.). First, 400 mL of a buffer (pH 7.0 containing K_2_HPO_4_ (1 g/L) and NaHCO_3_ (10 g/L) was evenly added to 400 g of the acid sulfate soil; the remaining buffer was removed before mixing in 180 g of the base organic material with 20 g of CaCO_3_. These experiments were performed in duplicate for the PSB and control (non-PSB) groups. For germination experiments, 40 tomato seeds were spread evenly in a pot and covered with approximately 1 cm of soil. Each PSB strain was inoculated and mass-cultured in 500 mL of R2A broth medium that was previously autoclaved at 121 °C for 15 min. The liquid culture was centrifuged at 8000 rpm for 30 min, the supernatant was removed, and the pellets were collected for experiments. Each strain pellet was washed with AFW and diluted so the OD_600_ was 0.23 to 0.24. A total of 5 mL of the suspension of each strain was diluted into 45 mL AFW, and then evenly dispensed to pots. The PSB inoculations were performed on day-0 and day-20, and colony forming units (CFUs) were measured. For these counts, bacteria were collected by centrifugation, suspended in 25 mL of a 0.85% NaCl solution and diluted to 10^−1^ to 10^−8^ CFUs, and then 100 μL was inoculated into an R2A agar plate. Strains were cultured for 4 days, and the number of colonies was counted. After inoculating the PSB, plants were maintained in a growth chamber (HB-301M-3, Hanbaek, Daejeon, Korea), with a 12 h light/12 h dark cycle at a temperature of 25 °C. In addition, 200 mL of distilled water was added every 2 days to correct for water loss from vaporization.

### 2.4. Analytical Methods

During the plant growth experiments, the pH, EC, and tATP of the soils, and the percentage germination and number of leaves per plant were measured in each pot. The changes in pH and EC were monitored using a multi-meter (Orion Star A215, Chelmsford, MA, USA) in water discharged from the bottom of the pots after adding 200 mL of distilled water every 2 days. The tATP (total-adenosine tri-phosphate = intra-cellular ATP + extra-cellular ATP) in soil from each replicate was measured using the Luminultra DSA-100 Kit (LuminUltra, Fredericton, NB, Canada). For these tATP measurements, soil samples were collected, placed in a tube containing an Ultralyse enzyme, pretreated, and transferred to an assay tube. The luminase enzyme was then added, followed by measurement using the PhotonMaster (LuminUltra, Fredericton, NB, Canada). Samples were collected for tATP measurements on days 1 and 50 after inoculating each strain. Means and standard deviations were obtained from three independent replicates. The germination rate and the number of total leaves were monitored after sowing and measured once every 2 to 3 days from day-25. All results were analyzed using IBM SPSS Statistics version 25 (IBM Corporation, Armonk, NY, USA). The comparison of means was performed using Duncan’s multiple range test.

## 3. Results and Discussion

### 3.1. Characterization of Isolated PSB Strains

We isolated two PSB strains from the roots of reed plants roots that were collected from a stream and performed 16S ribosomal RNA gene sequencing for species identification. The results indicated that one strain was in the genus *Caballeronia* and the other was in the genus *Methylobacterium*. We therefore named these isolates *Caballeronia* EK and *Methylobacterium* PS, and they were registered with NCBI under the accession number MK747357 (strain PS) and MK747358 (strain EK), respectively. We also performed phylogenetic analyses of these strains based on the 16S rRNA sequences (Figure 1).

Initially, we determined the optimal culture conditions and physiological properties of these two PSB strains (Table 3). *Caballeronia* sp. EK formed white colonies and grew at 10 to 35 °C (optimum: 25 °C); grew from pH 4.0 to 11.0 (optimum: pH 7.0); and grew at 0 to 3.5% NaCl concentration (optimum: 0% NaCl). *Methylobacterium* sp. PS formed pink colonies and grew at 15 to 35 °C (optimum growth: 30 °C); grew from pH 5.0 to 11.0 (optimum: pH 9.5); and grew at 0–6% NaCl concentration (optimum: 1% NaCl).

### 3.2. Physicochemical Changes of Soils during the Plant Growth Experiments

During the plant growth experiments, we measured the changes in pH and EC in the water discharged from the bottom of the pots after adding 200 mL of distilled water every 2 days (Figure 2). These measurements indicated that the pH changed markedly over time, but was similar in the three groups (within 0.4 pH units) at any single time. The pH increased soon after PSB inoculation in the EK and PS groups on day-0. The pH also increased in the control group due to the addition of organic supplement at that time. Eventually, the pH stabilized near 6.8, but it increased slightly after the second inoculation on day-20. The pH subsequently decreased as the effect of the organic supplement declined. However, the pH stabilized at approximately 6.0 to 6.2. We considered this pH to have no significant adverse effects on the growth of the microbes and tomato plants [22]. Similar to the timing of the changes in pH, the EC values tended to decrease soon after the initial inoculation on day-0, and there was a minor peak after the second inoculation on day-20, and the EC then stabilized near 1 mS/cm.

### 3.3. ATP-Based Assay of Total Bacterial Activity

We also measured tATP levels in the soils of the different treatment groups on day-1 and day-50 (Table 4). The soil tATP level is a quantitative indicator of the number of living cells [23]. These concentrations of tATP are expressed in terms of microbial equivalents (MEs)/g soil, as representative of the total number of viable microbes and assuming that all free ATP degraded quickly, and that detected ATP was only from living or recently decreased cells [23,24]. On day-1 (1 day after inoculation), soils inoculated with *Caballeronia* sp. EK had the highest tATP level (mean: 740,168 MEs/g soil), followed by *Methylobacterium* sp. PS. (692,184 MEs/g of soil). The controls had much lower levels of tATP (423,601 MEs/g soil), thus confirming that there were significantly more living microbes in the pots inoculated with PSB. However, the control pots still had living cells because of the addition of the organic supplement. We also measured soil tATP levels on day-50. The mean value was greatest in soils that received *Caballeronia* sp. EK inoculations (879,597 MEs/g soil), and this level was 18.9% greater than in soils that received *Methylobacterium* sp. PS inoculations (713,415 MEs/g soil) and 45.2% greater than the controls (482,273 MEs/g soil). Notably, from day-1 to day-50, the tATP level increased in all three groups. PSB strains had a positive effect on the growth of *Lycopersicon esculentum* L. in the acid sulfate soils, and it can be considered as the result explaining the role of microorganisms in promoting vegetation. Our results confirmed that the *Caballeronia* sp. EK strain provided the greatest increase in the number of soil microbes.

### 3.4. Germination and Leaf Growth

We measured the percentage of germination of *Lycopersicon esculentum* L. seeds on day-52 in the different treatment groups (Table 5). The results indicated that soils inoculated with *Caballeronia* sp. EK had 80% germination, much greater than soil inoculated with *Methylobacterium* sp. PS (57.5%) and the control soil (20%).

We measured the mean number of leaves of *Lycopersicon esculentum* L. plants in the different groups every 2 to 4 days beginning on day-25 (Table 6). These measurements did not include the cotyledons, and the values were from the average of 40 plants per pot. Within-group analysis indicated the control group had 8.7 leaves on day-25, and this number declined to 4.7 leaves on day-47. Over time, these plants had fewer leaves, but leaf size increased. The *Methylobacterium* sp. PS group had 18.3 leaves on day-25 (2.1-fold more than the control), and this number stayed at 17.3 leaves as the size of the leaves increased, and then declined after day-40 even though leaf size continued to increase. The *Caballeronia* sp. EK group had 10.3 leaves on day-25, but the number and size of these leaves increased, and the maximum number was 17.3 leaves from day-40 to day-52. These results indicated that soil inoculated with these two PSB strains led to increased germination and development of more leaves in tomato plants. Therefore, the PSB analyzed here may have potential for use as biofertilizers in the restoration of acid sulfate soils.

Many species of *Methylobacterium* occur in the rhizomes of plants, and they can secrete indole-3-acetic acid (IAA), a promoter of plant growth [25]. Many species of *Caballeronia* and some closely related genera can produce antibiotics or fix atmospheric nitrogen [26]. This may be why the two PSB strains used in this study promoted plant growth. Most chemical fertilizers are designed to provide nitrogen, because soils normally have abundant carbon and phosphorus [20]. However, inoculation of PSB can solubilize the inorganic phosphorus in soils, and provide a nearly permanent source of phosphorus to plants if these microbes persist [16,17]. The soils described here that were treated with PSB led to significantly increased germination and improved plant growth. These results suggest it may be possible to achieve efficient vegetation of acid sulfate soils by promotion of phosphorus absorption. The three major essential elements for plant growth are carbon, nitrogen, and phosphorus. Carbon is typically abundant, and occurs as carbon dioxide in the air. Nitrogen in the form of chemical salts is often readily available due to the activities of nitrogen fixing microbes in the soil [17]. However, most phosphorus exists in a form that cannot be absorbed by plants. Although less phosphorus than carbon and nitrogen is required for growth, phosphorus is a limiting factor for plant growth in most environments. Soil microbes are often related to plant growth. As such, the phosphate solubilizing activity of microorganisms in the rhizosphere plays an important role in plant growth. Organic acids in PSB solubilize phosphorus, in which the hydroxyl and carboxyl groups chelate the soil phosphorus, converting it into water-soluble forms [26]. These chelated forms of phosphorus can be taken up by plants, indicating that inoculation of PSBs may help to facilitate restoration of vegetation.

The *Methylobacterium* sp. PS strain examined in the present study is a typical genus of the Methylobacteriaceae family that occurs in plant stems, leaves, flowers, and roots, and usually has a pink carotenoid pigment. *Methylobacterium* species utilize C1 compounds as a carbon source, and can also use methanol generated from plant metabolism as a substrate. *Methylobacterium* species can reportedly use byproducts from plants and can invade plant seeds (including tomato seeds), thereby promoting the secretion of phytohormones. Previous research reported that plants treated with *Methylobacterium* had longer roots than control plants that did not receive this treatment [14,25,27]. Phytohormones play an important role in plant growth and yield. IAA, the first auxin to be isolated, is an important phytohormone that plays an important role in cell growth, division, and differentiation [25,27,28]. Because some bacteria can synthesize IAA, previous researchers suggested that inoculating soils with IAA-producing bacteria could stimulate plant growth. In fact, previous research indicated that the IAA level was greater in soils inoculated with *Methylobacterium* sp., as were the amounts of two cytokines (*trans*-zeatin riboside and dihydro-zeatin riboside) in tomato plants growing in these soils [14,25]. Moreover, the resulting plants had increased levels of IAA and cytokines, and greater growth controls [14,25,29].

The MBF1239 strain of *Methylobacterium* produces two types of alkyl quinolones, a class of compounds that have antibiotic properties. Many prescribed quinolone-based antibiotics are effective against Gram-negative bacteria (except *Pseudomonas aeruginosa*). Thus, some species in the *Methylobacterium* genus can be classified as pathogenic bacteria and others as antibiotic-producing bacteria. It was hypothesized that infection of plants by pathogenic bacteria or nematodes functioned as a stressor that led to the symbiosis of plants with antibiotic-producing bacteria [26,30]. Some of these microbes may also perform nitrogen fixation. These nitrogen-fixing bacteria use ATP to reduce atmospheric nitrogen to produce ammonia, and the ammonia is then used for intracellular synthesis of various amino acids, such as glutamine. Some bacteria can fix nitrogen alone, and other bacteria fix nitrogen by forming symbiotic relationships with plant roots [26,30]. For example, leguminous bacteria (*Rhizobia*) infect the roots of legumes, leading to the formation of root nodules, a symbiotic relationship. In this case, *Rhizobia* provide a nitrogen source to the plants, and the plants provide ions and a suitable habitat necessary for bacterial growth. Some species in the genus *Burkholderia* perform independent nitrogen fixation, and others in this genus perform symbiotic nitrogen fixation [31]. Further investigations should clarify how and to what extent inherent molecular properties of secreted hormones by PSB impact plant growth.

The acidic drainage from acidic sulfate soils (produced by the oxidation of pyrite) contains high concentrations of heavy metals, and this is the cause of the damage to farmland and the poor growth of vegetation. Remediation of acidic soils can be achieved by covering up the pyrite with alkaline material to reduce atmospheric exposure, and this intervention reduces acidic drainage and stabilizes the soil. However, this method is not an eco-friendly intervention for the restoration of poor landscapes. For the restoration of acidic soils, priority should be given to introducing vegetation, an eco-friendly intervention that stabilizes slopes and improves landscapes. When a slope is covered with plants, the organic content in the soil continuously increases over time, and this reduces the dissolved oxygen concentration of rainwater that penetrates the embankment layer. This effect reduces pyrite oxidation and thereby reduces the generation of acidic drainage. A general method to suppress acidic drainage is prevention of the oxidation of sulfide minerals by application of a surface coating. However, the materials responsible for acidic drainage still contain some amount of water, so acidic drainage will eventually increase over time, and this underground acidic drainage can move to the surface by capillary action, eventually killing plants. A more appropriate method for the efficient revegetation of acid sulfate soils is the application of biofertilizers using PSB, as described in the present study.

## 4. Conclusions

We isolated and characterized two PSB, *Methylobacterium* sp. PS and *Caballeronia* sp. EK, and determined the effects on the germination and growth of tomato plants when they were added as supplements to acid sulfate soils. Relative to untreated soil (control), soils treated with each bacterial strain led to 37.5% to 60% increased germination after 52 days, a 2- to 3-fold increased number of plant leaves after 52 days, and 18.9% to 45.2% increased soil tATP levels after 50 days. The two strains of PSB described here have potential for use as biofertilizers that promote vegetation growth in acid sulfate soils.

## Figures and Tables

**Figure 1 ijerph-18-08919-f001:**
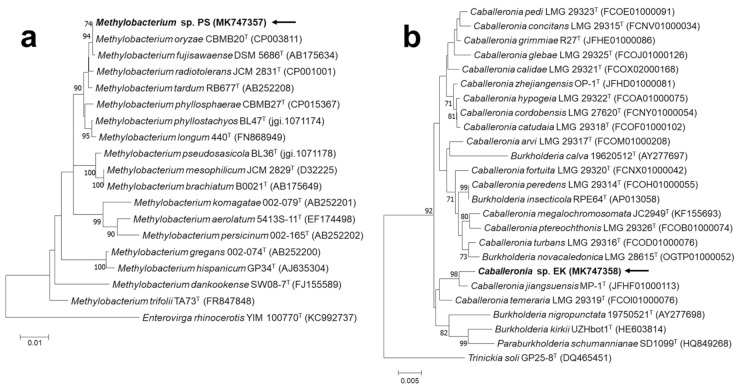
Neighbor-joining tree showing phylogenetic positions of (**a**) *Methylobacterium* sp. PS (MK747357) and (**b**) *Caballeronia* sp. EK (MK747358) with closely related strains based on 16S rRNA gene sequences, reconstructed by MEGA7. Bootstrap values of ≥70% are shown (1000 replicates).

**Figure 2 ijerph-18-08919-f002:**
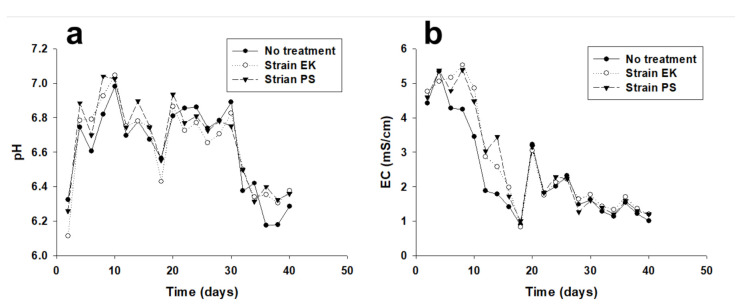
Changes in pH (**a**) and EC (**b**) in each treatment group during the plant growth experiments. Measurements were performed in duplicate replicates.

**Table 1 ijerph-18-08919-t001:** Characteristics of the acid sulfate soil.

Sand	Silt	Clay	pH	EC (ds/m)	CEC (cmol_c_/kg)	OM	C	N	Avail. *P*	Exchangeable Cations (cmol_c_/kg)				
%						%			mg/kg	Ca	Mg	K	Na	Al
52	18	30	3.4	8.97	7.03	2.68	4.29	0.39	38.96	1.63	0.49	1.37	0.15	4.25

Abbreviations: EC, electrical conductivity; CEC, cation exchange capacity; OM, organic matter.

**Table 2 ijerph-18-08919-t002:** Chemical properties of the organic soil supplement.

Chemical Property	Value
Moisture content (%)	45.93
pH	7.2
EC (ds/m)	0.84
Cation exchange capacity (cmol/kg)	35.98
Nitrogen (%)	0.68
OM (%)	26.83

**Table 3 ijerph-18-08919-t003:** Physiological properties of isolated phosphate solubilizing bacteria in this study.

Physiological Property	*Caballeronia* sp. EK	*Methylobacterium* sp. PS
NO_3_	+	+
Tryptophane	-	-
Glucose fermentation	-	-
*L*-argnine (ADH)	-	-
Urea (URE)	-	-
Esculine	+	+
Gelatine	+	+
4-nitrophenyl-*D*-galatctopyranoside	-	-
*D*-glucose (GLU)	+	+
*L*-arabinose	+	+
*D*-mannose	+	+
*D*-mannitol	+	+
*N*-acetyl-glucosamine	+	+
*D*-maltose	-	+
Potassium gluconate	+	+
Capric acid	+	-
Adipic acid	+	-
Malic acid	+	+
Trisodium citrate	+	-
Phenylacetic acid	+	-
Alkaline phosphatase	+	-
Esterase (C4)	+	+
Esterase Lipase (C8)	+	-
Lipase (C14)	-	-
Leucine arylamidase	+	-
Valine arylamidase	+	+
Crystine arylamidase	-	-
Trypsin	-	-
*α*-Chymotypsin	-	-
Acid phospatase	+	+
Naphtol-AS-Bi-phosphohydrolase	+	+
*α*-galatosidase	-	-
*β*-glucuronidase	-	-
*α*-glucosidase	-	-
*β*-glucosidase	-	-
*N*-acetyl-*β*-glucosaminidase	-	-
*α*-mannosidase	-	-
*α*-fucosidase	-	-

**Table 4 ijerph-18-08919-t004:** Soil tATP levels in the different treatment groups.

Treatment	tATP, Mean ± SD (MEs/g Soil) ^a^	
	Day-1	Day-50
No bacteria #1	399,335 ± 656	466,026 ± 702
No bacteria #2	447,866 ± 702	498,520 ± 691
*Methylobacterium* sp. PS #1	675,566 ± 713	704,328 ± 684
*Methylobacterium* sp. PS #2	708,801 ± 756	722,502 ± 711
*Caballeronia* sp. EK #1	732,855 ± 639	883,542 ± 706
*Caballeronia* sp. EK #2	747,481 ± 725	875,652 ± 723

^a^ Means and standard deviations were from three independent replicates. Measurements were performed 1 day and 50 days after PSB inoculation.

**Table 5 ijerph-18-08919-t005:** Germination of tomato seeds in the different treatment groups.

Treatment	Germinated Seeds (n/N) ^a^	Percent Germination
No bacteria—1	6/40	15%
No bacteria—2	10/40	25%
*Methylobacterium* sp. PS—1	22/40	55%
*Methylobacterium* sp. PS—2	24/40	60%
*Caballeronia* sp. EK—1	34/40	85%
*Caballeronia* sp. EK—2	30/40	75%

^a^ Measurements were performed after 52 days.

**Table 6 ijerph-18-08919-t006:** Number of leaves of *Lycopersicon esculentum* L. plants in the different treatment groups.

	Mean Number of Leaves on the Indicated Day *											
Treatment	Day 25	Day 27	Day 31	Day 33	Day 35	Day 38	Day 40	Day 42	Day 45	Day 47	Day 49	Day 52
No bacteria	8.7c	8.7c	8.7c	7.7c	7.7b	7.7b	5.7c	5.7c	5.7c	4.7c	4.7c	4.7c
*Methylobacterium* sp. PS	18.3a	17.3a	17.3a	17.3a	17.3a	17.3a	11.3b	11.3b	11.3b	11.3b	10.0b	10.0b
*Caballeronia* sp. EK	10.3b	15.3b	15.3b	15.3b	16.3a	16.3a	17.3a	17.3a	17.3a	17.3a	17.3a	17.3a

* Mean values with different letters in a single column are significantly different (Duncan’s test: *p* = 0.05).

## Data Availability

The data that support the findings of this study are available from the corresponding author, upon reasonable request.

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
