# Peer review of "Effect of Treating Acid Sulfate Soils with Phosphate Solubilizing Bacteria on Germination and Growth of Tomato (Lycopersicon esculentum L.)"

_ijerph, 2021, doi:10.3390/ijerph18178919_

Round 1
Reviewer 1 Report
The article talks about the isolation of two phosphate solubilizing bacteria that can be used as biofertilizers in acid sulfate soils. I suggest the following corrections.
- DNA extraction method is missing
- More details about PCR reactions
- Soil physicochemical methods are missing
- More details about organic soil supplement, is it forest soil or peat? due to the high organic matter.
- The redaction of plant experiments needs to be improved because it is confusing, for example, first talk about plants, then talk about inoculum, then talk about plants again, and after that talk about microorganisms. I don't understand why colony-forming units were measured if your inoculum were applied through optic density measured. Add more details about pot size and how many seeds per pot were applied? How many pots were used for the experiment? Please explain where soil supplement were used and in which proportion? how many days the plant experiments were done?
- Details of the formulation of the R2A medium are missed or the manufacturer.
- Table 1. Why the organic matter content is bigger than total carbon? Check this result. Please change N for total N and C for total C if applicable
- Results of isolates characterized could be on a Table for better understanding.
- Table 3. The values have SD, however, the methodology does not describe, how many measurements were done?
- Table 5. These results are from the average of 40 plants per isolated or 40 plants per pot? Please explain it in the text.
- The English style in line 93 needs to be corrected
- The following line 345 in the conclusions is not verified with the experiments carried out: improving the absorption of soluble phosphorus by plants.
- What is the purpose of organic supplement soil in the experiment?
- Add data about how isolates were selected based on phosphorus solubilizing activity.
Reviewer 2 Report
In the current manuscript by Jae Hwan Kim et al., two bacteria Methylobacterium sp. PS and Caballeronia sp. EK were isolated from roots of reed plants that growing near Daedeok stream in Korea. Authors tested the optimal conditions in which the two bacteria grow and found 25 °C, pH 7.0 and 0% NaCl are optimum for Caballeronia sp. EK while 30°C, pH 9.5 and 1% NaCl are optimum for Methylobacterium sp. PS. pH and electrical conductivity changes of soils treated with the two isolated bacteria were monitored. Authors also measured tATP levels in soil treated with isolated Methylobacterium sp. PS and Caballeronia sp. EK and found that these two bacteria increased tATP when compared with No bacteria control. Promotion on Lycopersicon esculentum L. seeds germination and leaf amount by treatment with Methylobacterium sp. PS and Caballeronia sp. EK were also observed by authors.
Authors stated that the two isolated bacteria Methylobacterium sp. PS and Caballeronia sp. EK are PSB (Phosphate solubilizing bacteria), but didn’t provide any evidence that indicates the two isolated bacteria have phosphate solubilizing ability. Thus data on this point should be added.
Minor comments are as follows:
- The two bacteria were isolated from plant roots that growing near Daedeok stream. Is the soil acid sulfate soil? What is pH for the soil near Daedeok stream? What CFU can the isolated bacteria reach in acidic soil and normal soil?
- In Table 1, abbreviations (EC, and OM) were used, but not in table 2. To be consistent, abbreviations should be used in table 2 as well.
- Line 137, abbreviation AFW was used in line 87 for the first time. So here AFW can be used directly.
- Line 177, Since authors have the data, why not include the data? ‘data not known’ can’t make the following statements convincing.
- Line 194, the last ‘comma’ should be replaced with ‘and’ because malic acid is the last listed.
- Figure 2, the values are mean values or representative values from different repeat? If mean values are used, please add error bars and clarify how many replicates were used.
- Table 3 and 4, please include statistical analysis. How many times were the experiments repeated?
- Line 265-337, discussion section is more like a review. Actually, discussion can be more relevant to results section. That means discuss based on results.
- Line 338, ‘conclusions’ section is following section 3 ‘Results and discussion’. So here should be 4.
Reviewer 3 Report
Firstly, I would recognize that I really liked the manuscript, especially the introduction and the well-structured discussion of the results. However, I would demand to the authors a clearer description of the material and methods. I have found some parts of this section very messy and hard to understand. I recommend a new description of the "plant growth experiments" section, especially the part where the authors explained the bacterial inoculation procedure.
Other minor suggestions would be:
- In my opinion Tables, 1 and 2 and Figure 2 could be placed as Supplemental Information, in order to improve the manuscript readability.
- In Table 3 and 4, I have found difficult to understand the meaning of the #1 and #2 division of the results, and furthermore, the text barely helps with it.
- Minor misspelling faults across the text like "every other day" instead of "every day"; or some word repetition like roots in line 169 or b-glucosidase in line 199.
Finally, I have some questions to the authors that I like to see resolved in the final manuscript:
- Why do you use bacterial inoculations with OD600 between 0.23-0.24? What is the biological reason?
- Why don´t you characterized the strains IAA and CKs production? Because most of your discussion is based on the Methylobacterium (also Caballeronia) effect through the hormone production. Maybe you could add these measures for the final manuscript, there are easy to implement and there are tons of bibliography to consult (some examples:
- Gilbert, S., Xu, J., Acosta, K., Poulev, A., Lebeis, S., and Lam, E. (2018). Bacterial production of indole related compounds reveals their role in association between duckweeds and endophytes. Front. Chem. 6:265. doi: 10.3389/fchem. 2018.00265
- De Hita D, Fuentes M, Zamarreño AM, Ruiz Y and Garcia-Mina JM (2020) Culturable Bacterial Endophytes From Sedimentary Humic Acid-Treated. Plants. Front. Plant Sci. 11:837. doi: 10.3389/fpls.2020.00837
- Lin, G. H., Chang, C. Y., and Lin, H. R. (2015). Systematic profiling of indole-3-acetic acid biosynthesis in bacteria using LC-MS/MS. J. Chromatogr. B Anal.Technol. Biomed. Life Sci. 988, 53–58. doi: 10.1016/j.jchromb.2015.02.025)
Round 2
Reviewer 2 Report
My comments were answered and manuscript was modified accordingly.